# Comparative immunogenicity and reactogenicity of heterologous ChAdOx1-nCoV-19-priming and BNT162b2 or mRNA-1273-boosting with homologous COVID-19 vaccine regimens

Verena Klemis[1], Tina Schmidt [1], David Schub [1], Janine Mihm[2,5], Stefanie Marx[1], Amina Abu-Omar[1], Laura Ziegler[1], Franziska Hielscher[1], Candida Guckelmus[1], Rebecca Urschel[1], Stefan Wagenpfeil[3], Sophie Schneitler[4], Sören L. Becker[4], Barbara C. Gärtner[4], Urban Sester[2,5] & Martina Sester [1] ✉

Comparative analyses of the immunogenicity and reactogenicity of homologous and heterologous SARS-CoV-2 vaccine-regimens will inform optimized vaccine strategies. Here we analyze the humoral and cellular immune response following heterologous and homologous vaccination strategies in a convenience cohort of 331 healthy individuals. All regimens induce immunity to the vaccine antigen. Immunity after vaccination with ChAdOx1-nCoV-19 followed by either BNT162b2 ($n = 66$) or mRNA-1273 ($n = 101$) is equivalent to or more pronounced than homologous mRNA-regimens ($n = 43$ BNT162b2, $n = 59$ mRNA-1273) or homologous ChAdOx1-nCoV-19 vaccination ($n = 62$). We note highest levels of spike-specific CD8 T-cells following both heterologous regimens. Among mRNA-containing combinations, spike-specific CD4 T-cell levels in regimens including mRNA-1273 are higher than respective combinations with BNT162b2. Polyfunctional T-cell levels are highest in regimens based on ChAdOx1-nCoV-19-priming. All five regimens are well tolerated with most pronounced reactogenicity upon ChAdOx1-nCoV-19-priming, and ChAdOx1-nCoV-19/mRNA-1273-boosting. In conclusion, we present comparative analyses of immunogenicity and reactogenicity for heterologous vector/mRNA-boosting and homologous mRNA-regimens.

Three COVID-19 vaccines, the ChAdOx1-nCoV-19 vector-based vaccine (ChAdOx) and the two mRNA vaccines BNT162b2 (BNT) or mRNA-1273, are authorized as homologous dual-dose regimens and are widely used in Europe and the USA. In addition, heterologous combinations of vector vaccines followed by boosting with either of the two mRNA vaccines are recommended in some parts of Europe including Germany[1,2].

[1]Department of Transplant and Infection Immunology, Saarland University, Homburg, Germany. [2]Department of Internal Medicine IV, Saarland University, Homburg, Germany. [3]Institute for Medical Statistics, Epidemiology and Medical Informatics, Saarland University, Campus Homburg/Saar, Homburg, Germany. [4]Institute of Medical Microbiology and Hygiene, Saarland University, 66421 Homburg, Germany. [5]Present address: SHG Kliniken, Völklingen, Germany. ✉e-mail: martina.sester@uks.eu

We and others have recently shown that the heterologous ChAdOx/BNT mRNA vaccine combination elicited similar antibody- and CD4 T-cell levels as the homologous BNT regimen, and levels in both regimens were higher than after homologous ChAdOx vaccination[3–7]. Moreover, SARS-CoV-2-specific CD8 T-cell levels after heterologous vaccination were significantly higher than the homologous regimens including either ChAdOx or BNT. Finally, antibodies elicited after heterologous ChAdOx/BNT were shown to have neutralizing activity against the SARS-CoV-2 wild type as well as variants of concern including Delta[8,9].

Homologous combinations of the three vaccines have shown remarkable ability to prevent SARS-CoV-2 infection and COVID-19 disease[10–12], with differences in efficacy and effectiveness between the vaccines, especially between vector-based and mRNA-based compounds. Whether this is associated with differences in immunogenicity is poorly studied due to the lack of head-to-head studies. Moreover, standardized assays to estimate correlates of protection have only recently emerged to allow a more detailed comparison of immunogenicity[13,14]. A Spanish study on ChAdOx-BNT compared to ChAdOx alone, and the Com-COV study involving all combinations of the ChAdOx and BNT vaccines were the first randomized trials to show strong immunogenicity of a ChAdOx-BNT prime-boost regimen[6,7]. This was confirmed in real-world immunogenicity studies on heterologous vector/mRNA vaccine combinations, although most studies have initially focused on combinations including BNT[3–6]. The more recent randomized Com-COV-2 study also included a ChAdOx-mRNA-1273 group and confirmed higher antibody and T-cell levels compared to homologous boosting[15]. One additional study has shown that heterologous boosting with mRNA-1273 after ChAdOx-priming induced higher levels of antibodies than the homologous vector regimen, but cellular immunity induced by this combination was not assessed in parallel[16]. Together this suggests that heterologous mix-and-match regimens offer similar or more pronounced immunogenicity as homologous mRNA regimens, but head-to-head analyses on all homologous and heterologous combinations of authorized dual-vaccine combinations ChAdOx, BNT, and mRNA-1273 are currently lacking. We therefore prospectively enrolled a convenience cohort of immunocompetent individuals to study the immunogenicity and reactogenicity of the three homologous and two heterologous combinations of these vaccines. This included analyses of SARS-CoV-2-specific antibodies and neutralizing capacity as well as specific CD4 and CD8 T cells and their functional characteristics.

## Results

### Study population

The study was conducted among 331 healthy individuals mainly including health care personnel at Saarland University Medical Center, who either received homologous regimens with ChAdOx ($n = 62$), BNT ($n = 43$), or mRNA-1273 ($n = 59$), or heterologous vaccinations with ChAdOx-priming followed by a boost with either the BNT ($n = 66$) or the mRNA-1273 vaccine ($n = 101$) (Fig. 1 and Table 1). Despite no known history of SARS-CoV-2 infection, one female was positive for nucleocapsid-specific IgG, and was excluded from further analysis. Due to convenience sampling based on current recommendations, the mean time between the two vaccinations was shorter for the homologous mRNA regimens ($5.7 \pm 0.7$ weeks) as compared to the vector-based regimens ($11.9 \pm 0.9$ weeks). In addition, the group showed some differences in age and gender (Table 1). Blood sampling was carried out at a median of 14 (IQR 2) days after the second vaccination. In differential blood counts, leukocyte and granulocyte numbers differed between the groups with the highest numbers found after homologous mRNA-1273 vaccination. The numbers of monocytes, lymphocytes, and lymphocyte subpopulations such as B cells, CD4, and CD8 T cells did not differ. Among B cells, plasmablast numbers, which were identified as CD38 positive cells among IgD⁻CD27⁺ CD19-positive switched-memory B cells were also highest in individuals after homologous mRNA-1273 vaccination (Table 1).

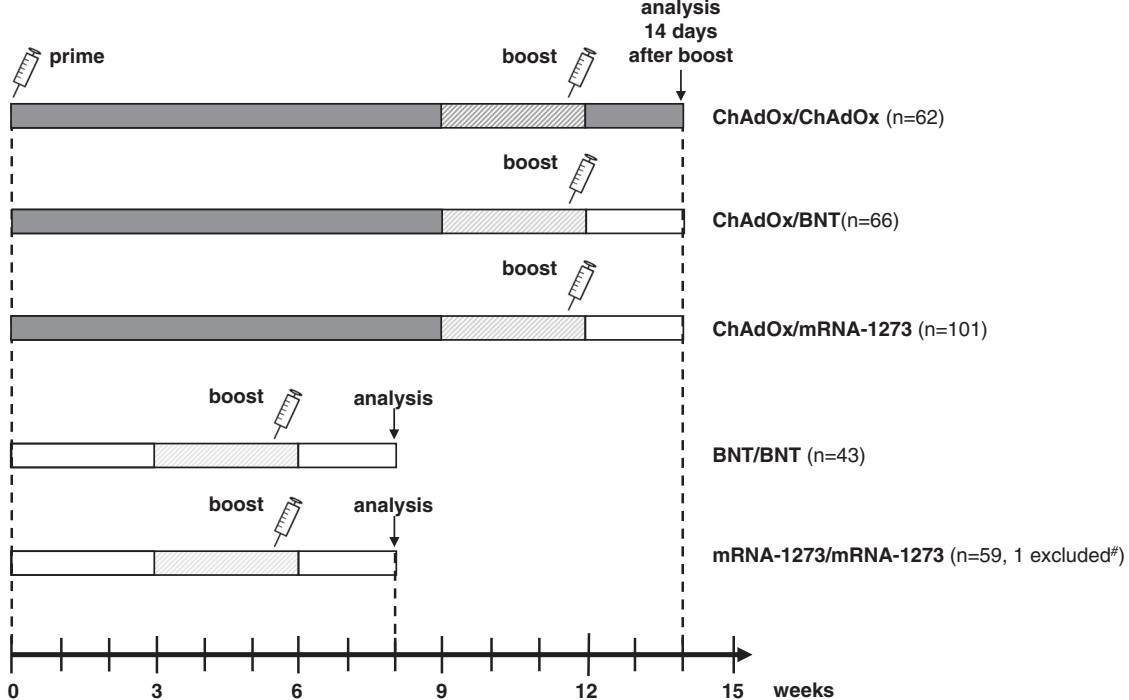

**Fig. 1 | Design of the study.** Schematic representation of the five vaccine regimens (three homologous: ChAdOx/ChAdOx $n = 62$, BNT/BNT $n = 43$, mRNA-1273/mRNA-1273 $n = 59$; two heterologous: ChAdOx/BNT $n = 66$, ChAdOx/mRNA-1273 $n = 101$). Shown are the time frames between the first (prime) and the second (boost) vaccination, and between the boost vaccination and the day of blood analysis. #One individual of the mRNA-1273/mRNA-1273 group was excluded from further analysis due to detectable IgG towards the SARS-CoV-2 nucleocapsid.

**Table 1 | Demographic and clinical characteristics of the study population**

| 1° vaccine<br>2° vaccine | ChAdOx<br>ChAdOx[1] | ChAdOx<br>BNT[2] | ChAdOx<br>mRNA-1273[3] | BNT<br>BNT | mRNA-1273<br>mRNA-1273 | |
|---|---|---|---|---|---|---|
| | $n = 62$ | $n = 66$ | $n = 101$ | $n = 43$ | $n = 58$ | *p* value |
| Years of age (mean ± SD) | 52.5 ± 10.9 | 45.0 ± 10.9 | 38.2 ± 13.7 | 52.9 ± 18.7 | 41.9 ± 15.2 | <0.0001§ |
| Female gender, *n* (%) | 41 (66.1) | 53 (80.3) | 82 (81.2) | 26 (60.5) | 38 (65.5) | 0.020† |
| Weeks between 1° and 2° vaccination, (mean ± SD) | 11.9 ± 1.1 | 11.7 ± 1.0 | 12.0 ± 0.7 | 5.4 ± 1.0 | 5.9 ± 0.2 | |
| Analysis time [days after 2° vaccination], median (IQR) | 14 (2.25) | 14 (1) | 14 (1) | 14 (2) | 15 (2) | |
| Differential blood cell counts | $n = 62$ | $n = 65$ | $n = 101$ | $n = 41$ | $n = 57$ | |
| Leukocytes (cells/μl), median (IQR) | 7000 (2400) | 6400 (2000) | 6800 (1850) | 6100 (2300) | 7700 (2550) | 0.018† |
| Granulocytes (cells/μl), median (IQR) | 4014 (1715) | 3856 (1391) | 4100 (1363) | 3611 (2176) | 4617 (1981) | 0.006† |
| Monocytes (cells/μl), median (IQR) | 561 (267) | 546 (233) | 531 (173) | 520 (217) | 568 (176) | 0.211† |
| Lymphocytes (cells/μl), median (IQR) | 2174 (1045) | 2103 (899) | 2170 (828) | 2189 (786) | 2241 (1127) | 0.720† |
| CD3 T cells (cells/μl), median (IQR)# | 1527 (817) | 1524 (735) | 1498 (696) | 1567 (663) | 1636 (890) | 0.840† |
| CD4 T cells (cells/μl), median (IQR) # | 919 (645) | 962 (413) | 937 (448) | 1031 (526) | 1066 (627) | 0.863† |
| CD8 T cells (cells/μl), median (IQR) # | 383 (236) | 385 (271) | 396 (241) | 340 (259) | 388 (266) | 0.448† |
| CD19 B cells (cells/μl), median (IQR)# | 204 (145) | 205 (135) | 202 (126) | 189 (168) | 244 (148) | 0.529† |
| Plasmablasts (cells/μl), median (IQR)# | 0.478 (0.696) | 0.483 (0.722) | 0.528 (0.473) | 0.471 (0.476) | 0.868 (0.785) | 0.001† |

[1]Refers to ChAdOx1-nCoV-19 by AstraZeneca; [2]Refers to BNT162b2 by BioNTech/Pfizer, [3]Refers to mRNA-1273 by Moderna; #B and T-cell counts were calculated on 61 ChAdOx-ChAdOx, 64 ChAdOx-BNT, 101 ChAdOx-mRNA-1273, 39 BNT-BNT, and 57 mRNA-1273-mRNA-1273 vaccinated individuals, respectively. §Ordinary one-way ANOVA with Tukey´s multiple comparisons test. †$X^2$ test; †two-sided Kruskal–Wallis test; source data are provided as a Source Data file.

## Differential induction of antibodies and T cells after homologous and heterologous vaccination

Spike-specific IgG was detectable in all individuals, but their levels were significantly higher in individuals boosted with mRNA vaccines as compared to individuals after homologous ChAdOx vaccination (Fig. 2a, $p < 0.0001$). When comparing heterologous regimens, boosting with mRNA-1273 led to numerically higher IgG levels (6043 (IQR 4396) BAU/ml) than boosting with BNT (4275 (IQR 4080) BAU/ml). Likewise, among homologous regimens, IgG levels were higher after mRNA-1273 vaccination (5529 (IQR 5755) BAU/ml) than in BNT vaccinated individuals (3438 (IQR 3287) BAU/ml), although the differences did not reach statistical significance. As with IgG levels, neutralizing inhibitory capacity of spike-specific antibodies determined using a surrogate assay was high and reached a maximum of 100% in the majority of mRNA-boosted individuals, which contrasted with significantly lower neutralizing activity after homologous ChAdOx vaccination (median 77.8% (IQR 33.5%), $p < 0.0001$, Fig. 2a).

Vaccine-induced CD4 and CD8 T cells were quantified after stimulation with overlapping peptides encompassing the spike protein. Activation-induced T cells were identified based on CD69 and IFNγ, TNFα, and IL-2. A representative example of CD69-positive spike-specific CD4 and CD8 T cells producing IFNγ from a 49-year-old female after the second homologous mRNA-1273 vaccination is shown in Supplementary Fig. 1, and data from all individuals analyzed after the second vaccination are summarized in Fig. 2b. Spike-specific CD4 T-cell levels in the homologous ChAdOx vaccine group were significantly lower than in all other groups. Among mRNA-boosted regimens, median levels of spike-specific CD4 T cells were highest after heterologous ChAdOx1/mRNA-1273 vaccination (0.29% (IQR 0.23%)). Not only did a boost with mRNA-1273 outperform

heterologous boosting with BNT after ChAdOx-priming (0.18% (IQR 0.17%), $p < 0.01$), but CD4 T-cell levels were also higher after homologous vaccination with mRNA-1273 (0.24% (IQR 0.27%) than with BNT (0.10% (IQR 0.08%), $p < 0.0001$). Interestingly, the two heterologous regimens also led to a strong induction of spike-specific CD8 T cells (0.29% (IQR 0.57%) for BNT and 0.40% (IQR 0.60%) for mRNA-1273), with significantly higher levels than all three homologous regimens (Fig. 2b, $p < 0.0001$). All vaccine-induced effects on CD4 and CD8 T cells were specific, as no differences in *Staphylococcus aureus* Enterotoxin B (SEB)-reactive CD4 and CD8 T cells were observed between the five groups (Fig. 2c). Finally, in line with a pronounced induction of vaccine-induced T cells, CTLA-4 expression was strongly induced on spike-specific CD4 and CD8 T cells of all individuals after heterologous vaccination and in both homologous mRNA regimens, whereas CTLA-4 expression on specific T cells after homologous ChAdOx vaccination was significantly lower (Fig. 2d). These differences in CTLA-4 expression were also spike-specific, as CTLA-4 expression on SEB-reactive CD4 and CD8 T cells were similarly low in all five groups (Fig. 2e).

When analyzing correlations between spike-specific IgG levels, neutralizing activity, and spike-specific CD4 and CD8 T cells (Fig. 2f and Supplementary Table 1), neutralizing activity showed a strong correlation with IgG levels in each vaccine subgroup. Likewise, spike-specific CD4 and CD8 T cells showed a significant correlation. In line with the previous findings[3], CD4 T cells correlated with IgG in ChAdOx/ChAdOx and ChAdOx/BNT vaccinated individuals only. In addition, it is interesting to note that IgG levels correlated with CD8 T-cell levels in the three homologous vaccine groups only, whereas no such correlation was found for the two heterologous vaccine groups, which may be a result of the exceptionally high CD8 T-cell response in these two groups (see Fig. 2b).

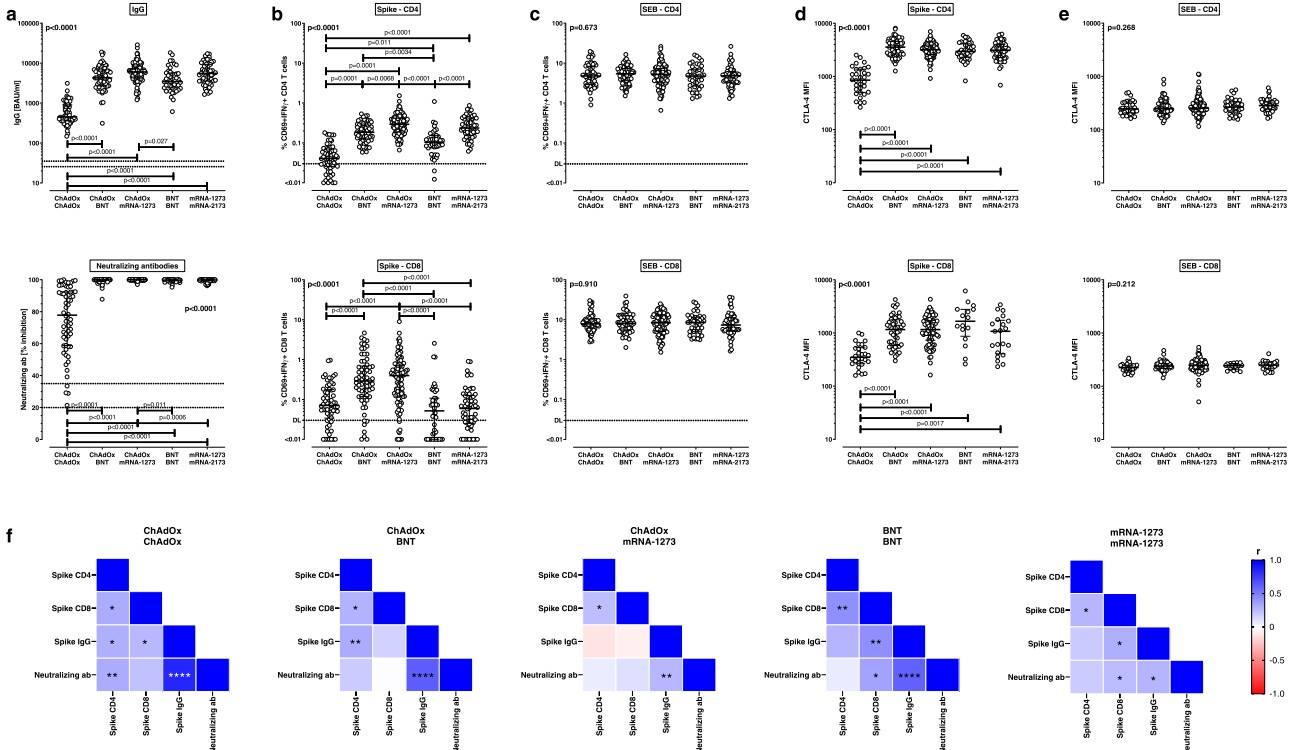

**Fig. 2 | Antibody and T-cell responses against the SARS-CoV-2 spike protein after homologous COVID-19 vaccine regimens or heterologous ChAdOx-priming and BNT- or mRNA-1273-boosting.** Cellular and humoral immune parameters were analyzed 13–18 days post vaccination and compared between individuals with different homologous or heterologous COVID-19 vaccine regimens: homologous ChAdOx vaccination (*n* = 62), heterologous ChAdOx/BNT vaccination (*n* = 66), heterologous ChAdOx/mRNA-1273-vaccination (*n* = 101), homologous BNT vaccination (*n* = 43) or homologous mRNA-1273-vaccination (*n* = 58). **a** ELISA and surrogate neutralization assays were performed to quantify levels of spike-specific IgG and neutralizing antibodies. Intracellular cytokine staining after antigen-specific stimulation of whole blood samples allowed for flow-cytometrical determination of SARS-CoV-2 spike-specific (**b**) and SEB-reactive (**c**) CD4 and CD8 T-cell levels. Reactive cells were identified by co-expression of CD69 and IFNγ

among CD4 or CD8 T cells and subtraction of reactivity of respective negative control stimulations. CTLA-4 expression was determined on **d** spike-specific and **e** SEB-reactive CD4 and CD8 T cells in all samples with at least 20 cytokine-positive CD4 and CD8 T cells. **f** Correlation matrix of spike-specific T-cell and antibody responses among each group. Bars in **a**–**e** represent medians with interquartile ranges. Differences between the groups were calculated using two-sided Kruskal–Wallis test with Dunn´s multiple comparisons post-test. Correlations in **f** were analyzed according to two-tailed Spearman (see also Supplementary Table 1). Dotted lines indicate detection limits for antibodies in **a**, indicating negative, intermediate, and positive levels or levels of inhibition, respectively as per manufacturer's instructions, and detection limits for SARS-CoV-2-specific CD4 T cells in **b** and **c**. Source data are provided as a Source Data file. IFN Interferon, MFI median fluorescence intensitiy, SEB *Staphylococcus aureus* enterotoxin B.

As the five groups differed in age and gender due to convenience sampling and recruitment according to national recommendations (Table 1), a subgroup analysis was performed among 40 individuals per vaccination regimen which were matched for age and gender (Supplementary Table 2). As shown in Supplementary Fig. 2, between-group differences in IgG levels, neutralizing activity and spike-specific T cells largely remain the same. In the whole cohort, adjusting for age and gender as confounders in a non-parametric regression analysis showed that both confounders did not have any significant effect on immunological parameters (Supplementary Table 3). When testing for interactions of age within each vaccine group with the homologous ChAdOx group as a reference, age had no effect on T-cell levels and neutralizing antibody activity; the only effect of age was found for IgG levels within each of the two homologous regimens (*p* = 0.003 for BNT/BNT and *p* = 0.015 for mRNA-1273/mRNA-1273, Supplementary Table 3).

Based on national recommendations, the interval between the first and the second dose was longer for ChAdOx-primed groups than for individuals on homologous mRNA regimens (see Table 1). If within-group comparisons were restricted to regimens with the same interval, differences between the respective groups remain the same as those indicated in Fig. 2.

## Functional differences in vaccine-induced T cells after homologous and heterologous vaccination

Apart from IFNγ, we also analyzed spike-specific induction of the cytokines TNFα and IL-2. As with IFNγ, differences between the groups were similar for CD4 T cells producing TNFα or IL-2 (Fig. 3a, b), or for cells producing any of the three cytokines alone or in combination (Fig. 3c). This also held true for spike-specific CD8 T cells, except for IL-2 producing CD8 T cells, where levels were generally lower and only showed subtle differences between the groups (Fig. 3b). To assess functionality on a single cell level, cytokine profiles of spike-specific CD4 and CD8 T cells were characterized after Boolean gating (Supplementary Fig. 3). This allowed distinction of seven subpopulations including polyfunctional cells simultaneously expressing all three cytokines, two cytokines or one cytokine only (Fig. 4). The cytokine-expression profiles showed significant differences between the vaccine regimens, and the highest percentage of polyfunctional CD4 T cells was observed for the three vector-primed regimens. These three regimens also showed the highest percentage of CD8 T cells expressing IFNγ and TNFα, which was the dominant fraction among spike-specific CD8 T cells (Fig. 4a). The differences in cytokine-expression profiles were spike-specific, as SEB-reactive cytokine expression did not differ among the groups (Fig. 4b).

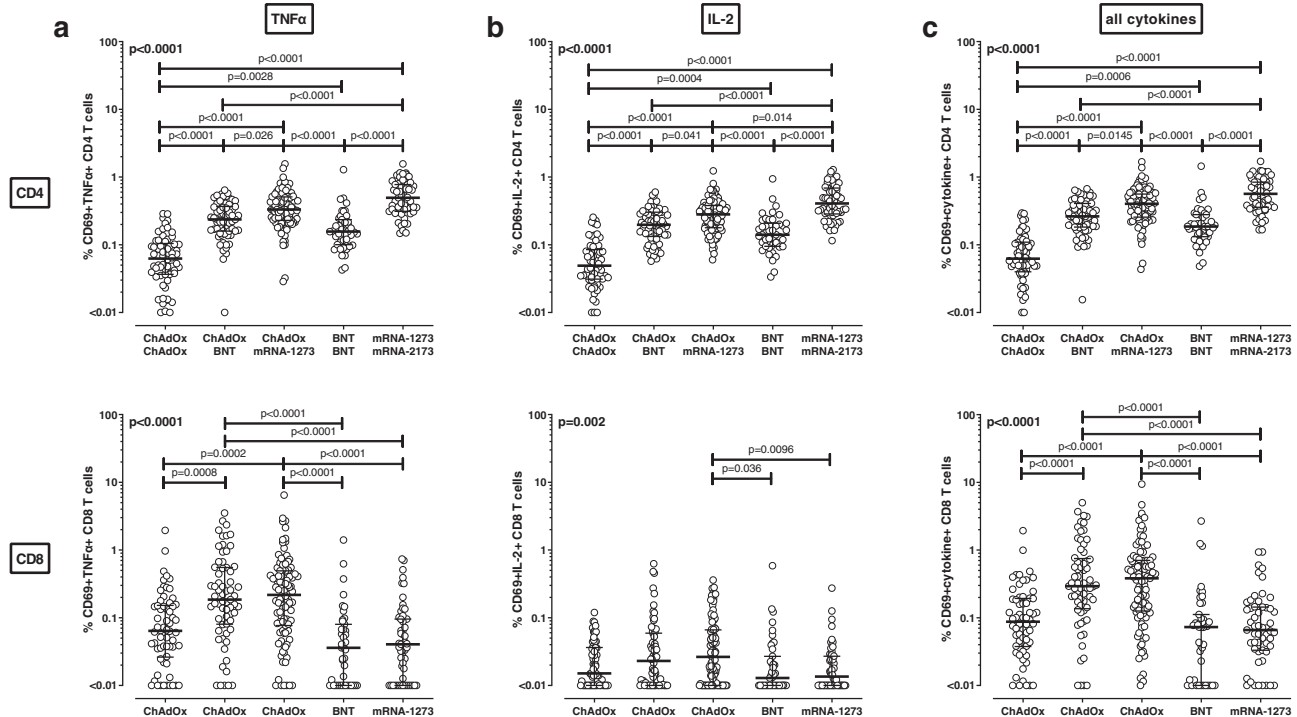

**Fig. 3 | SARS-CoV-2-specific cytokine expression after homologous or heterologous COVID-19 vaccination.** Levels of TNFα and IL-2-expressing T cells and combined expression of either of the cytokine IFNγ, TNFα and/or IL-2 were compared between individuals who either received homologous ChAdOx vaccination (*n* = 62), heterologous ChAdOx/BNT vaccination (*n* = 66), heterologous ChAdOx/mRNA-1273-vaccination (*n* = 101), homologous BNT vaccination (*n* = 43) or homologous mRNA-1273 vaccination (*n* = 58). Percentages of CD69+ TNFα+ (**a**), CD69+ IL-2+ **b** or CD69-positive cells co-expressing at least one of the cytokines TNFα, IL-2, or IFNγ (**c**) among total CD4 (upper panel) or CD8 T cells (lower panel) were determined after stimulation of whole blood samples with overlapping peptides of SARS-CoV-2 spike protein and subtraction of background reactivity from negative control stimulations. Bars represent medians with interquartile ranges and two-sided Kruskal–Wallis test with Dunn´s multiple comparisons post-test was used to calculate differences between the groups. Source data are provided as a Source Data file. IFN interferon, IL interleukin, SEB *Staphylococcus aureus* enterotoxin B, TNF tumor necrosis factor.

## Differences in reactogenicity after homologous and heterologous vaccination

Local and systemic adverse events within the first week after the first and the second vaccination were self-recorded using a questionnaire (Fig. 5 and Supplementary Tables 4 and 5). Irrespective of the vaccine type, local adverse events such as pain at the injection site were reported with similar frequency in individuals after the first vaccination. Swelling at the injection site was overall less frequently observed with the lowest percentage among BNT-primed individuals (Fig. 5b). Systemic adverse events including fever, headache, fatigue, chills, gastrointestinal manifestations, myalgia, and arthralgia after priming were most frequent in individuals after ChAdOx vaccination, which also was associated with more frequent use of antipyretic medication (Fig. 5c and Supplementary Table 4). After the second vaccination, local adverse events were least frequent after homologous ChAdOx vaccination, and most frequent in both heterologous and in the homologous mRNA-1273 regimens. The occurrence of systemic adverse events clearly dominated in individuals after heterologous boosting with mRNA-1273, followed by homologous mRNA-1273 vaccination and heterologous BNT-boosting (Fig. 5a, c, Supplementary Table 5). Individual perception of severity was scored higher after secondary vaccination in both homologous mRNA regimens (Fig. 5d). In contrast, more than 75% of subjects after both the homologous ChAdOx and heterologous BNT vaccination were more affected by the primary vaccination with the vector. Despite the strong reactogenicity after vector-priming, it was interesting to note that a sizable fraction of subjects after heterologous boosting with mRNA-1273 was more severely affected by the secondary vaccination, which contrasts with observations in the heterologous BNT vaccine group. Likewise, among individuals after homologous vaccination, the second vaccination with

mRNA-1273 was more frequently perceived as more severe, although this vaccine was already strongly reactogenic after the primary vaccination. Overall, it therefore appeared that both the homologous and the heterologous regimens that included BNT were better tolerated than the respective mRNA-1273 regimens.

## Discussion

The three vaccines ChAdOx1-nCoV-19, BNT152b2, and mRNA-1273 were developed and authorized to be administered as a homologous prime/boost regimen. Our study now provides detailed head-to-head immunogenicity and reactogenicity data comparing all three homologous COVID-19 vaccine regimens with heterologous combinations of ChAdOx-priming followed by either BNT- or mRNA-1273-boosting. We show that all regimens are immunogenic, but show considerable differences in the extent of vaccine-induced antibody and T-cell responses. The most striking finding was that immunogenicity of the mRNA-1273 containing regimens was more pronounced than the respective BNT vaccine combinations, which held true for both the homologous and the heterologous regimens. Correspondingly, homologous or heterologous boosting with mRNA-1273 was less well tolerated as compared to the other regimens.

We and others have shown that heterologous ChAdOx/mRNA regimens led to a strong induction of antibodies and T cells[3–7], whereas antibody levels after heterologous vaccination with BNT followed by ChAdOx were lower[6]. Up to now, immunogenicity of heterologous regimens was so far mainly studied in individuals primed with ChAdOx followed by boosting with the BNT mRNA. Moreover, most studies did not differentiate between specific CD4 and CD8 T cells. We now show that heterologous boosting with mRNA-1273 led to similar antibody and T-cell response patterns with a particular strong induction of CD8

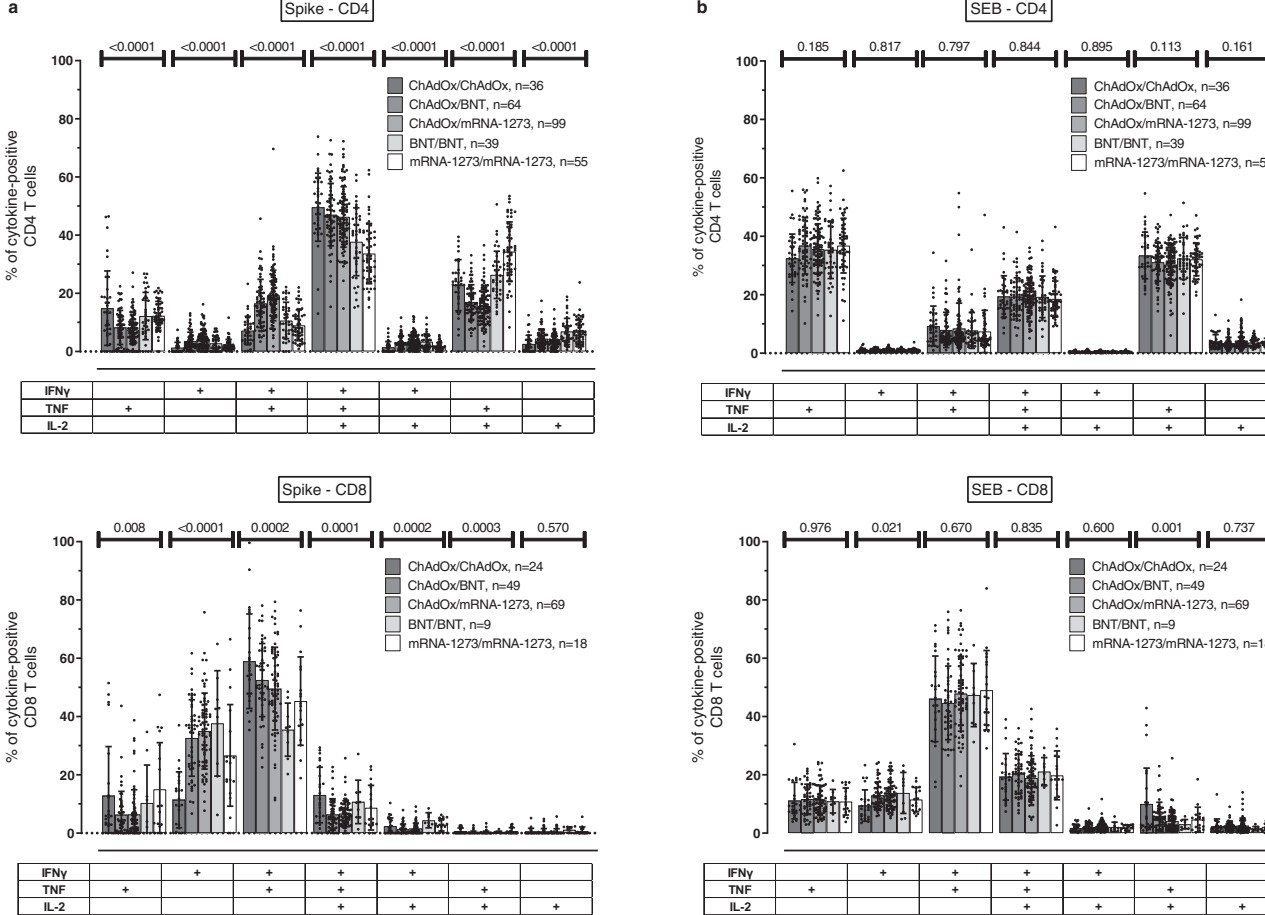

**Fig. 4 | Antigen-specific cytokine-expression profiles of T cells in individuals with different homologous and heterologous COVID-19 vaccination regimens.** After antigen-specific stimulation (**a**) or polyclonal stimulation with *Staphylococcus aureus* enterotoxin B (SEB, **b**) of whole blood samples from individuals with different homologous or heterologous vaccination regimens, cytokine-expressing CD4 and CD8 T cells were subclassified into seven subpopulations according to single or combined expression of IFNγ, IL-2, and TNFα. Blood samples from all individuals were analyzed. To ensure robust statistics, only samples with at least 30 cytokine-expressing CD4 or CD8 T cells after normalization to the negative control stimulation were considered (with the number of samples in each vaccine group indicated in the figures). Bars in **a** and **b** represent means and standard deviations, and ordinary one-way ANOVA tests were performed. Source data are provided as a Source Data file. IFN interferon, IL interleukin, TNF tumor necrosis factor.

T cells. When comparing the two mRNA vaccines, immunogenicity was generally more pronounced after boosting with mRNA-1273 in both the heterologous and homologous vaccine group. This may be related to a higher dosage of the mRNA (100 μg vs. 30 μg) and/or different formulations of lipid nanoparticles[11,12]. Our data confirm that a secondary vaccination with the vector is less potent in boosting antibodies and T cells as compared to all mRNA-containing regimens[3–7]. As this may be related to preformed or induced neutralizing immunity towards the vector backbone[17], the boosting effect is enhanced using either heterologous combinations or homologous regimens with mRNA vaccines that use lipid nanoparticle vaccine carriers. Interestingly, despite poor immunogenicity after homologous ChAdOx-boosting, all three ChAdOx-primed regimens led to the highest percentage of multifunctional T cells upon secondary boosting, which may result from the potent ability of the ChAdOx vector for T-cell priming[18]. In general, spike-specific CD8 T cells known to mediate protection from severe COVID-19[19] were most strongly induced after heterologous boosting. Based on the fact that CD8 T cells did not correlate with antibody levels in the two heterologous vaccine groups, analysis of antibodies alone may be insufficient to evaluate protection from severe disease in these groups. We have previously shown that a pronounced induction of antigen-specific T-cell levels after infection with SARS-CoV-2[20], with varicella zoster virus[21], or after influenza vaccination[22] is paralleled by an upregulation of CTLA-4 on specific T cells which may serve to counteract excessive T-cell proliferation and/or T-cell mediated immunopathology. Interestingly, we now show that the highest expression of CTLA-4 on spike-specific CD4 and CD8 T cells was found in the four vaccine groups with the most pronounced induction of CD4 and CD8 T cells after vaccination, whereas CTLA-4 expression in individuals after homologous ChAdOx vaccination was significantly lower, which supports a less potent boost of T-cell immunity. On the B-cell side, stronger immunogenicity of the heterologous as compared to the homologous BNT regimen was recently found to be associated with a higher percentage of spike-specific activated memory B cells[23]. While this finding may result from a more pronounced T-cell help, this may also explain the higher avidity[4] and the higher neutralizing capacity[23] of antibodies observed after heterologous boosting. Moreover, the presentation of the same antigen in two different vaccine formulations may differentially affect antigen presentation and trigger immunity towards immunodominant epitopes from different angles, which may influence avidity.

The three homologous regimens have shown remarkable but variable efficacy and effectiveness regarding protection from infection. The differences in immunogenicity between the homologous regimens largely correspond with similar differences in efficacy in the range of 70.4% for ChAdOx[10], 95% for BNT2[12], and 94.1% for mRNA-1273[11]. Similar differences among the three homologous vaccine regimens were also reported for effectiveness in nationwide observational

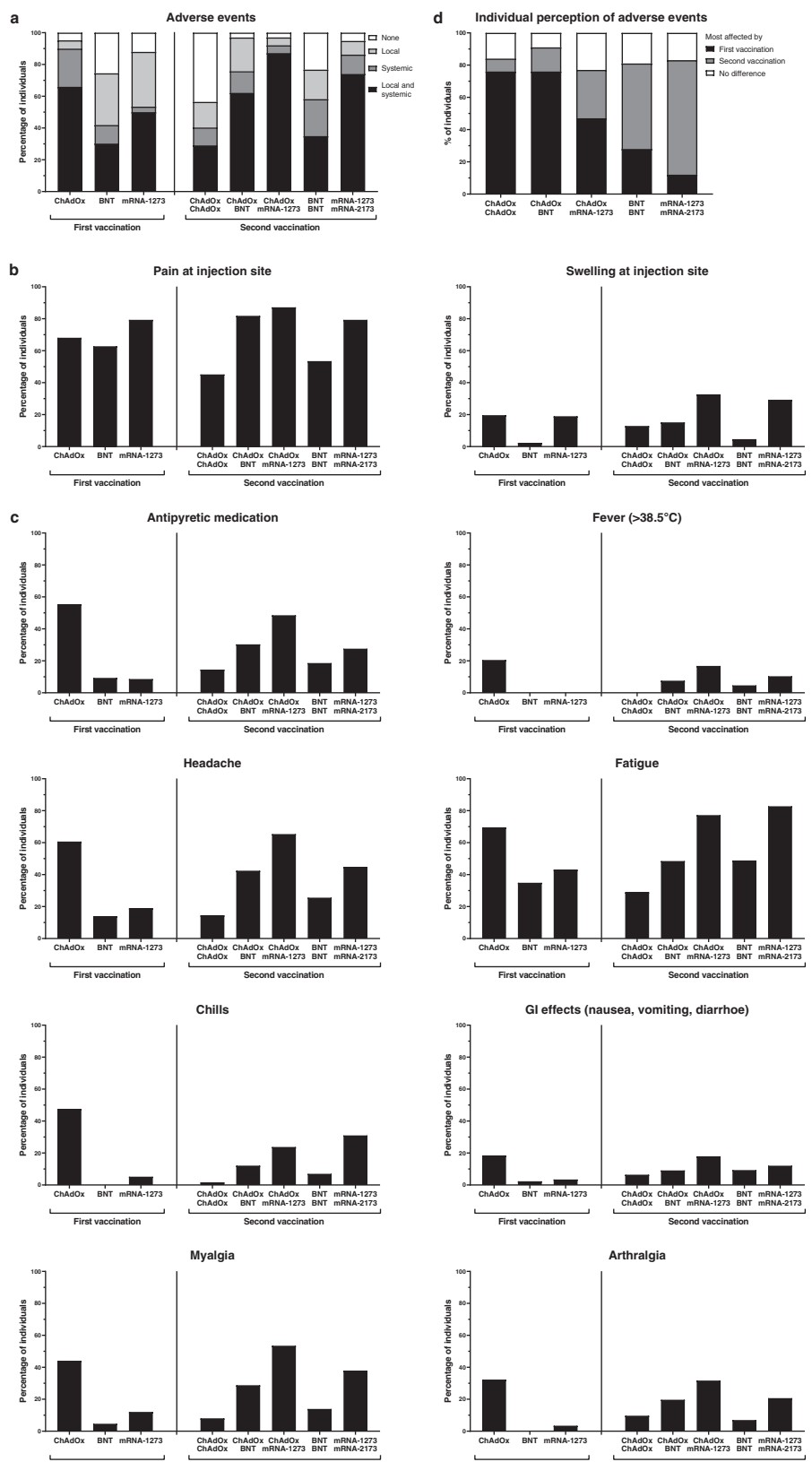

**Fig. 5 | Reactogenicity after primary and secondary vaccination with homologous and heterologous COVID-19 vaccine regimens.** According to their COVID-19 vaccine regimens, individuals were classified into three groups after dose 1 (ChAdOx vector (*n* = 229), BNT (*n* = 43) or mRNA-1273 vaccine (*n* = 58)) and five groups after dose 2 (homologous: ChAdOx/ChAdOx, *n* = 62; BNT/BNT, *n* = 43; mRNA-1273/mRNA-1273, *n* = 58; heterologous: ChAdOx/BNT, *n* = 66; ChAdOx/mRNA-1273, *n* = 101). Self-reported reactogenicity within the first week after each vaccine dose was assessed using a standardized questionnaire. The presence of local or systemic adverse events in general (**a**), substantial local (**b**) or systemic adverse events (**c**), and individual perception of which of the two vaccinations affected more (**d**) are shown. Statistical analyses of differences between the groups after the first and the second vaccination are shown in Supplementary Tables 2 and 3. Source data are provided as a Source Data file.

studies[24–29], whereas real-world effectiveness data for heterologous regimens had been limited. In this regard, a recent nationwide cohort study in ChAdOx-primed individuals from Sweden found effectiveness of 50% after homologous boosting, 67% after BNT-boosting and the highest effectiveness of 79% after boost with mRNA-1273[30]. This indicates that our observation of a higher immunogenicity of the ChAdOx/mRNA-1273 regimen may also translate into a higher effectiveness. In addition, in line with an equivalent or more pronounced immunogenicity of heterologous mRNA-boosting, a study from France provided evidence for a higher effectiveness of the ChAdOx/BNT regimen as compared to homologous BNT vaccination[23]. Finally, the ChAdOx/BNT regimen in a Danish nationwide study reached a remarkable effectiveness of 88%, although no control groups with other regimens were analyzed in parallel[31]. While this direct comparison of immunogenicity and efficacy or effectiveness is intriguing, it needs to be emphasized that outcome definitions of infection and disease, as well as follow-up times, differ in the various trials. In addition, apart from immunogenicity, other factors may impact efficacy such as the populations studied, pre-existing immunity, and circulating strains of the virus.

The strength of our study is the large head-to-head analysis of immunogenicity and reactogenicity of five recommended two-dose homologous and heterologous vaccine combinations available in Germany and many other European countries in a real-world setting. Our study is limited by convenience sampling in a non-randomized study design, where study participants were enrolled according to national recommendations. Although this led to some differences in age within the five groups, between-group differences of immunological parameters among age-matched subgroups remained largely the same. Within a given regimen, an effect of age was only observed for IgG levels within the homologous mRNA groups. Therefore, at least in our cohort of individuals mainly including health care workers, age is unlikely to have a strong confounding impact on our results. Based on national recommendations, the interval between the first and the second vaccination was longer for the ChAdOx-primed groups (9–12 weeks) as compared to the homologous mRNA vaccine groups (3–6 weeks). This may represent another limitation, as immunogenicity and efficacy was previously shown to be higher with longer intervals[32,33]. Thus, although different spacing may influence immunogenicity in general, this did not account for the striking differences in immune responses among the three ChAdOx-primed groups, which had the same time interval between priming and boosting. Moreover, we observed clear differences in immunogenicity within the two homologous and the two heterologous regimens containing mRNA vaccines, although the respective intervals between priming and boosting were similar. A further limitation is a fact that we do not have any information on neutralizing activity towards variants of concern. However, the observational study from France showed that the heterologous ChAdOx/BNT regimen had enhanced activity towards the Delta variant as compared to homologous BNT vaccination[23].

Knowledge of the differences in immunogenicity and reactogenicity of homologous and heterologous vaccine combinations is of increasing importance for clinical practice. First, mixing different vaccine principles in heterologous vaccination regimens is already implemented for regular COVID-19 vaccination procedures in many countries due to the frequent occurrence of rare, but serious adverse events after ChAdOx-priming[34,35]. In general, all tested regimens were well tolerated, although the more pronounced immunogenicity in individuals boosted with mRNA-1273 was associated with a higher percentage of individuals with local and systemic adverse events. Second, vaccine shortage in many countries may necessitate the use of heterologous combinations to ensure broad vaccine coverage. In this regard, randomized and observational trials have now already been extended to other vaccine combinations for both dual-dose and booster regimens[15,36,37]. Finally, serial use of heterologous combinations of different vaccines is of increasing importance to optimize immunogenicity of a single dose or poorly immunogenic homologous regimens. As illustrated by the favorable immunogenicity of heterologous regimens in solid organ transplant recipients[18], this is of particular relevance for immunocompromised patients who exhibit a severely impaired immunogenicity after regular homologous vaccination; as reactogenicity is less of a concern in immunocompromised patients[18], mix-and-match regimens may offer the most favorable risk-benefit ratio for this population. Finally, as with other widely used vaccines, deviation from homologous series may become common practice for booster vaccinations after waning of vaccine-induced protection.

## Methods

### Study design and subjects

Our study complies with all relevant ethical regulations. The study was approved by the ethics committee of the Ärztekammer des Saarlandes (reference 76/20), and all individuals gave written informed consent. Study participants with no known history of SARS-CoV-2 infection were enrolled in this observational study prior to their secondary vaccination as described before[3]. Healthy immunocompetent volunteers (mainly employees at Saarland University Medical campus) were invited to participate in the study. No compensation was provided for participation. We enrolled participants on all five possible authorized dual-dose vaccine combinations as per recommendations in Germany including homologous regimens with ChAdOx, BNT or mRNA-1273, or heterologous regimens with ChAdOx-priming and boosting with either BNT or mRNA-1273 (Fig. 1)[1]. The choice of regimen, including the time interval between the first and the second vaccination (3-6 weeks for homologous mRNA regimens, and 9-12 weeks for all regimens with ChAdOx-priming) was based on recommendations[38] and not determined by the study. Study participants were enrolled prior to the second and in part prior to the first vaccination, and received a questionnaire for self-reporting of local and systemic adverse events within the first week after the first and second vaccination. Blood samples were collected during an interval of 13-18 days after secondary vaccination to determine lymphocyte subpopulations and SARS-CoV-2-specific humoral and cellular immunity. Primary vaccinations were performed between 18th of January and 10th of June 2021. Thirty-six individuals (12 ChAdOx/ChAdOx, 22 ChAdOx/BNT, 1 ChAdOx/mRNA-1273, 1 BNT/BNT) were enrolled in a separate observational study (SaarTxVac study). Their results on induction of humoral and cellular immunity were part of a subgroup of 70 immunocompetent individuals to comparatively study vaccine-responses in transplant recipients[18].

### Quantification of lymphocyte populations and plasmablasts

T cells, B cells and plasmablasts were quantified from 100 µl heparinized whole blood exactly as described before using monoclonal antibodies towards CD3 (clone SK7, final dilution 1:25), CD19 (clone HIB19, 1:40), CD27 (clone L128, 1:200), CD38 (clone HB7, 1:20) and IgD (clone IA6-2, 1:33.3). T and B cells were identified among total lymphocytes by expression of CD3 and CD19, respectively. CD4 and CD8 T cells were quantified after staining of CD4 (clone SK3, 1:100) and CD8 (clone RPA-T8). Plasmablasts were defined by the expression of CD38 among IgD-CD27 + CD19 positive switched-memory B cells. Antibodies are listed in Supplementary Table 4. Analysis was performed on a BD FACSLyric flow-cytometer and BD FACSuite software v1.4.0.7047 followed by data analysis using FlowJo software 10.6.2. Analyses of T cells, B cells, and plasmablasts were performed using a gating strategy as described before[3]. Absolute lymphocyte numbers were calculated based on differential blood counts.

## Quantification of vaccine-induced SARS-CoV-2-specific T cells

SARS-CoV-2-specific T cells were determined from heparinized whole blood after a 6h-stimulation with overlapping peptides spanning the SARS-CoV-2 spike protein (N-terminal receptor binding domain and C-terminal portion including the transmembrane domain, each peptide 2 μg/ml; JPT, Berlin, Germany) exactly as described previously[3,20]. Stimulations with 0.64% DMSO and with 2.5 μg/ml of *Staphylococcus aureus* enterotoxin B (SEB; Sigma) served as negative and positive controls, respectively. All stimulations were carried out in presence of co-stimulatory antibodies against CD28 and CD49d (clone L293 and clone 9F10, 1 μg/ml each). Immunostaining was performed using anti-CD4 (clone SK3, 1:33.3), anti-CD8 (clone SK1, 1:12.5), anti-CD69 (clone L78, 1:33.3), anti-IFNγ (clone 4 S.B3, 1:100), anti-IL-2 (clone MQ1-17H12, 1:12.5), anti-TNFα (clone MAb11, 1:20), and anti-CTLA-4 (clone BNI3, 1:50) and analyzed using flow-cytometry (BD FACS Canto II including BD FACSDiva software 6.1.3). Antibodies are listed in Supplementary Table 6. SARS-CoV-2-reactive CD4 or CD8 T cells were identified as activated CD69-positive T cells producing IFNγ. Moreover, co-expression of IL-2 and TNFα was analyzed to characterize cytokine-expression profiles using a gating strategy as described in Supplementary Fig. 3). Reactive CD4 and CD8 T-cell levels after control stimulations were subtracted from levels obtained after SARS-CoV-2-specific stimulation, and 0.03% of reactive T cells was set as detection limit as described before[3].

## Determination of SARS-CoV-2-specific antibodies and neutralization capacity

All antibody tests were performed according to the manufacturer´s instructions (Euroimmun, Lübeck, Germany) as described before[3]. SARS-CoV-2-specific IgG antibodies towards the receptor binding domain of SARS-CoV-2 spike protein were quantified using an enzyme-linked immunosorbent assay (ELISA, SARS-CoV-2-QuantiVac). Antibody binding units (BAU/ml) <25.6 were scored negative, ≥25.6 and <35.2 were scored intermediate, and ≥35.2 were scored positive. SARS-CoV-2-specific IgG towards the nucleocapsid (N) protein were quantified using the anti-SARS-CoV-2-NCP-ELISA. A neutralization assay based on antibody-mediated inhibition of soluble ACE2 binding to the plate-bound S1 receptor binding domain (SARS-CoV-2-NeutraLISA) was used at a single serum dilution. Surrogate neutralizing capacity was calculated as a percentage of inhibition (IH) by 1 minus the ratio of the extinction of the respective sample and the extinction of the blank value. IH < 20% was scored negative, IH ≥ 20 and <35 intermediate, and IH ≥ 35% positive.

## Statistics and reproducibility

No statistical method was used to predetermine sample size, as our study relied on convenience sampling and on size estimations from a previous study[3]. The individuals were recruited on the various regimens without randomization. During sample processing and analysis of primary data, the investigators were blinded to vaccine group allocation. Kruskal–Wallis test, followed by Dunn's multiple comparisons test, was performed to compare unpaired non-parametric data between groups (lymphocyte subpopulations, T-cell and antibody levels, CTLA-4 expression). Data with normal distribution were analyzed using ordinary one-way ANOVA (cytokine-expression profiles, age). Categorial analyses on gender and adverse events were performed using $X^2$ test. Correlations between levels of T cells, antibodies, and neutralizing activities were analyzed according to Spearman. A *p* value <0.05 was considered statistically significant. Analysis was carried out using GraphPad Prism 9.0 software (GraphPad, San Diego, CA, USA) using two-tailed tests. SPSS V27 including an R 3.6 plug-in for non-parametric regression analysis was used to determine the effect of confounders on SARS-CoV-2-specific immunity as described before[39].

## Reporting summary

Further information on research design is available in the Nature Research Reporting Summary linked to this article.

## Data availability

Figures 2–5, Table 1, Supplementary Figure 2, and Supplementary Tables 1–5 have associated raw data. Source data are provided with this paper, and data are available in a public repository as a single file, where an assignment of raw data for each figure or table is indicated (https://doi.org/10.5281/zenodo.6591908). As age may be subject to confidentiality, data in the repository refer to age groups.

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

## Acknowledgements

The authors thank Dr. Christina Baum and the team of the occupational health care center at Saarland University Medical Center for their support in enrolling participants. The authors also thank all participants in this study.

## Author contributions

V.K., T.S., D.S., U.S., B.C.G., S.S., and M.S. designed the study; V.K., T.S., D.S., U.S., and M.S. designed the experiments, V.K., S.M., F.H., A.A.-O., L.Z., C.G., R.U., and T.S. performed experiments; S.S., B.C.G., J.M., L.Z., S.L.B., and the U.S. contributed to study design, patient recruitment, and clinical data acquisition. T.S., M.S., U.S., and S.W. performed the statistical analysis. V.K., T.S., D.S., U.S., J.M., and M.S. supervised all parts of the study, performed analyses, and wrote the manuscript. All authors approved the final version of the manuscript.

## Funding

Financial support was provided by the State chancellery of the Saarland to M.S. Open Access funding was enabled and organized by Projekt DEAL.

## Competing interests

M.S. has received grant support from Astellas and Biotest to the organization Saarland University outside the submitted work, and honoraria for lectures from Biotest and Novartis. All other authors of this manuscript have no conflicts of interest to disclose.
