## [Peer Review File · Nature Communications]

Comparative immunogenicity and reactogenicity of heterologous ChAdOx1 nCoV-19-priming and BNT162b2 or mRNA-1273-boosting with homologous COVID-19 vaccine regimensREVIEWER COMMENTS

Reviewer #1 (Remarks to the Author):

Major Concerns

While these data are of interest there are other data already published from randomised controlled trials that describe the reactogenicity and immunogenicity for these regimens (and more). These published trials have consistent intervals between prime and boost. The data described in the current publication are inconsistent across prime-boost intervals which significantly and negatively impacts the importance of this study. This limitation means the immunogenicity reported across the different regimens should not be directly compared.

The authors fall into the common trap of using superior and ranking throughout the manuscript, this is not helpful in a global setting where vaccines are still limited in LMIC. Please consider the impact this language has in settings (LMIC) where RNA vaccines are in short supply, as this type of hierarchy results in vaccine hesitancy and low vaccine uptake of other recommended vaccines that are available and can protect individuals from severe disease and hospitalisation.

Minor concerns

Line 34 and throughout. Please ensure all blanket statements relating to comparative immunogenicity clearly states what time point the comparison refers to (e.g. add a qualifier stating the differences measured are at a 14 day boost time point)

Line 38 What dose, prime or boost, was the most pronounced reactogenicity measured at?

Line 41 – define notable

Line 52 – define superior

Line 58 – Globally there are more recommended vaccines e.g. the WHO has recommended a number of vaccines not described or referenced in the current publication

Line 60 – there are a number of publications defining correlates of protection, please address

Line 62 & 64 and throughout – there are randomised controlled trials that have already published these data, these must be included

Study population – only samples from regimens with the same prime boost interval should be included for analysis

Line 112 as with comment on line 34

Line 125 expand on the Tfh population induced after vaccination

Line 133 – line 135 Compared with 224 – these statements seem contradictory? Previous work in field demonstrates a relationship between CD4 and IgG – please expand on why this isn't measured here

Line 167 – Reactogenicity is reported after first and second dose, please do the same for immunogenicity

Line 187 – change the word perception

Line 197 – ‘the first’ this is untrue

Line 208 – boosting with BNT mRNA – untrue, other reports are published with mixed heterologous regimens

Line 237 & 238 – what is driving the higher avidity, higher neuts ?

Line 240 & 241 – the inference that immunogenicity alone drives efficacy considering the references cited is ill-advised. The authors should also consider that the definitions of disease in different trial protocols, as well as different population studied, pre-existing immunity and previous circulating strains all need to be considered as key drivers that will impact efficacy

Line 242 – 254 A number of different effectiveness values are discussed - are all of these values against the same (e.g. defined in the same manner to allow this type of comparison?)

Line 256 – ‘all recommended’ – untrue. WHO has recommended many more vaccines.

Reviewer #2 (Remarks to the Author):

This is a well-written and very interesting study comparing homologous vs heterologous prime/boost regimens of COVID-19 vaccines. The data are clearly presented in the figures. The text descriptions are also clear and match up with what is graphically presented. The conclusions are supported by the data.

Minor comments

1. A formal analysis should be performed to determine whether or not the differences in reactogenicity are significant. Visually there appears to be much more reactogenicity with heterologous boosting.

2. Following up on #1, the Discussion has little to no discussion about the differences in reactogenicity. This would be useful to put the findings into perspective. There is clearly improved immunogenicity but there may be a price to pay for this effect.

RESPONSES TO REVIEWER COMMENTS

REVIEWER COMMENTS

Reviewer #1 (Remarks to the Author)

Major Concerns

While these data are of interest there are other data already published from randomized controlled trials that describe the reactogenicity and immunogenicity for these regimens (and more). These published trials have consistent intervals between prime and boost. The data described in the current publication are inconsistent across prime-boost intervals which significantly and negatively impacts the importance of this study. This limitation means the immunogenicity reported across the different regimens should not be directly compared.

Regarding randomized studies: The reviewer is correct in pointing out that our study was not based on a randomized controlled study design, but rather represents a convenience cohort where vaccine spacing for the five various vaccine combinations followed German recommendations. This is now emphasized more clearly in the introduction on **p. 4/5** and in the discussion on **p. 12** and **16**. We have cited and discussed randomized controlled trials with similar combinations (of which one was published in 2022 after submission of our manuscript in the beginning of November), but we are not aware of other head-to-head studies comparing CD4 and CD8 T cells as well as humoral immunity among five groups of individuals including two different heterologous ChadOx-mRNA combinations and the three corresponding homologous regimens within one study.

Regarding vaccine intervals (see also response to the editor above): We thank the reviewer for this comment. We highlight that vaccine spacing followed German recommendations and differed between individuals primed with ChAdOx (ChAdOx/ChAdOx vs. ChAdOx/BNT and ChAdOx/mRNA-1273) and with the two homologous mRNA vaccine groups. Nevertheless, we feel that the comparative analysis of all groups as shown in figure 1 is of interest as it provides real world information on the humoral and cellular immunity after vaccination with all prime-boost regimens including vaccines commonly applied in Germany (and other countries).

In the manuscript, the differences in the intervals are transparently highlighted and discussed as follows: differences are stated in the methods section (**p. 17**), and were shown on **p. 6** and in table 1. In response to this comment, we have now also specifically addressed comparisons of regimes with the same intervals in the results section (**p. 9**). Our data show that immunogenicity differs even within regimens with the same interval (i.e. ChAdOx/ChAdOx vs. ChAdOx/BNT vs. ChAdOx/mRNA-1273 or the two homologous mRNA regimens), which indicates that the vaccine combination itself affects immunogenicity. Given the evidence on the additional effect of vaccine spacing in general, we have also included a sentence in the limitations section of the discussion on the impact of the time interval between priming and boosting (**p. 15**). We also added references, including a recent study on the impact of spacing for BNT regimens (Hall et al. Nat Immunol. 2022).

The authors fall into the common trap of using superior and ranking throughout the manuscript, this is not helpful in a global setting where vaccines are still limited in LMIC. Please consider the impact this language has in settings (LMIC) where RNA vaccines are in short supply, as this type

of hierarchy results in vaccine hesitancy and low vaccine uptake of other recommended vaccines that are available and can protect individuals from severe disease and hospitalization.

We thank the reviewer for raising this issue. As all regimens showed remarkable immunogenicity, we did not intend to cause vaccine hesitancy. We changed our wording accordingly throughout the manuscript and focused on an objective description of the differences between regimens.

Minor concerns

Line 34 and throughout. Please ensure all blanket statements relating to comparative immunogenicity clearly states what time point the comparison refers to (e.g. add a qualifier stating the differences measured are at a 14 day boost time point).

All immunological analyses refer to the second vaccination. In general, we allowed a time interval of 13-18 days after vaccination for analyses in our study design (methods section, **p. 17**). The median times after vaccination including interquartile ranges for each group are given in **table 1**. The fact that analysis was performed after the second vaccination (boosting) has now been stated more clearly throughout the manuscript (i.e. **p. 3, p. 7**).

Line 38 What dose, prime or boost, was the most pronounced reactogenicity measured at?

Reactogenicity after priming was most pronounced in individuals vaccinated with ChAdOx. The most pronounced reactogenicity after boosting was found in ChAdOx/mRNA-1273-vaccinated individuals. This was now phrased more clearly in the abstract (**p. 3**) and a more detailed comparison is found in the results section, where we have now also included a statistical analysis of the differences in reactogenicity as suggested by Reviewer 2 (**p. 10** and **supplementary tables 2 and 3**).

Line 41 – define notable

Including a more detailed definition and description of the differences would have exceeded the word count in the abstract. Therefore, the adjective “notable” was deleted in the last sentence of the abstract (and an objective description on the differences is included in the results section).

Line 52 – define superior

This referred to higher antibody- and T-cell levels. This was now added and wording was changed from “superior” to “higher” (**p. 4**).

Line 58 – Globally there are more recommended vaccines e.g. the WHO has recommended a number of vaccines not described or referenced in the current publication

In line 58, we referred to the three vaccines ChAdOx1 nCoV-19, BNT152b2 and mRNA-1273, which are the vaccines that were included in the current study. This was now emphasized more clearly to avoid misunderstanding (**p. 4**).

Line 60 – there are a number of publications defining correlates of protection, please address

We have now included publications dealing with correlates of protection. However, they were only recently emerging and not widely used in comparative analyses on different vaccine regimens to confirm results from effectiveness studies on the same regimens. This is the context in which this is now cited on **p. 4**.

Line 62 & 64 and throughout – there are randomised controlled trials that have already published these data, these must be included

The Com-COV study and the more recent Com-COV2 study were now included as landmark studies with a randomized study design. These studies were now included in the introduction (p. 4/5) and also in the last paragraph of the discussion (along with randomized trials including other vaccines, p. 16). Moreover, we emphasized more clearly that this paragraph was restricted to the vaccines that were licensed in our country as dual dose vaccine regimens at the time of the study.

Study population – only samples from regimens with the same prime boost interval should be included for analysis

Please see our response to this comment above.

Line 112 as with comment on line 34

We have now made clearer that we refer to analyses after secondary vaccination (p. 7). This had been described in the methods section on p. 17.

Line 125 expand on the Tfh population induced after vaccination

Please note that we have not analysed any Tfh cells, which would be CD4 T cells positive for CXCR5, or would be identified as IL-21 producing cells after stimulation. We therefore would prefer not to discuss this aspect in the results section.

Line 133 – line 135 Compared with 224 – these statements seem contradictory?

We thank the reviewer for pointing out that phrasing was misunderstanding. In the statement in line 224 we refer to the lack of correlation in the two heterologous vaccine groups which has now been clarified (p. 13).

Previous work in field demonstrates a relationship between CD4 and IgG – please expand on why this isn't measured here

Correlation coefficients between CD4 and IgG are shown in figure 1f and listed in detail in supplementary table 1, but a significant correlation was only found for two subgroups which were both primed with the ChAdOx vaccine (ChAdOx/ChAdOx and ChAdOx/BNT). The same observation was made in an independent study group in our previous publication (Schmidt et al. Nat Medicine). A sentence has now been included in the results section (p. 8). Interestingly, as with IgG and CD8 T cells in both heterologous vaccine groups, no correlation between IgG and CD4 T cells was found in ChAdOx/mRNA-1273 vaccinated individuals (see figure 1f), which may relate to significantly higher levels of CD4 T cells as compared to the ChAdOx/BNT group. As other studies on COVID-19 vaccine regimens (i.e. the Com-COV/Com-COV2 studies) have also found only weak correlations between IgG and T cells for some regimens, these results emphasize differential effects on T cells and antibodies among the various vaccine regimens which warrant further study.

Line 167 – Reactogenicity is reported after first and second dose, please do the same for immunogenicity

As described in the study design, immunogenicity data were only analyzed after the second vaccination. A comparative analysis of the first and the second dose was the focus of another independent smaller study previously published (Schmidt et al. Am J Transplant 2021). This study showed that antibody levels

were higher after mRNA priming whereas T-cell induction was more pronounced after ChAdOx vector priming (which was included in the discussion on **p. 13**).

Line 187 – change the word perception

This was replaced by observation (**p. 11**).

Line 197 – ‘the first’ this is untrue

This was now replaced and phrasing was altered, although we are not aware of any other study, where exactly these combinations were compared in a head-to-head study (see also next comment, **p. 12**).

Line 208 – boosting with BNT mRNA – untrue, other reports are published with mixed heterologous regimens

We now revised wording to be more accurate and also highlighted other reports with mixed heterologous regimens. In the context of the discussion, we would like to emphasize that we refer to authorized combinations of the ChAdOx vector and the two mRNA vaccines BNT and mRNA-1273 (**p. 12**).

Line 237 & 238 – what is driving the higher avidity, higher neuts?

Apart from a more pronounced T cell help in this vaccine combination, other potential aspects were not addressed in our study and therefore would be speculative. It may seem plausible that presentation of the same antigen in two different vaccine formulations may differentially affect antigen presentation and trigger immunity towards immunodominant epitopes from different angles, which may influence avidity (now discussed on **p. 13/14**).

Line 240 & 241 – the inference that immunogenicity alone drives efficacy considering the references cited is ill-advised. The authors should also consider that the definitions of disease in different trial protocols, as well as different population studied, pre-existing immunity and previous circulating strains all need to be considered as key drivers that will impact efficacy

We thank the reviewer for this comment and have added some more factors that may drive efficacy or effectiveness as suggested (**p. 14**, see also next comment).

Line 242 – 254 A number of different effectiveness values are discussed - are all of these values against the same (e.g. defined in the same manner to allow this type of comparison?)

The numbers that are given correspond to the respective values from the pivotal trials or effectiveness studies with differences in the outcome definitions. This was emphasized more clearly in the discussion on **p. 14** (see also previous comment).

Line 256 – ‘all recommended’ – untrue. WHO has recommended many more vaccines.

We realize that phrasing was misunderstanding. This was now clarified to refer to regimens recommended and available in our country and in many other European countries (**p. 15**).

Reviewer #2

This is a well-written and very interesting study comparing homologous vs heterologous prime/boost regimens of COVID-19 vaccines. The data are clearly presented in the figures. The

text descriptions are also clear and match up with what is graphically presented. The conclusions are supported by the data.

We thank the reviewer for this positive feedback.

Minor comments

1. A formal analysis should be performed to determine whether or not the differences in reactogenicity are significant. Visually there appears to be much more reactogenicity with heterologous boosting.

We have now prepared two tables summarizing detailed statistical analyses of differences in reactogenicity of the regimens after the first and the second vaccination (**p. 10**). These **supplementary tables 2** and **3** include a detailed list of p-values (based on Fisher's exact tests and on X^2 tests) for all adverse events shown in figure 3.

2. Following up on #1, the Discussion has little to no discussion about the differences in reactogenicity. This would be useful to put the findings into perspective. There is clearly improved immunogenicity but there may be a price to pay for this effect.

This was now included in the discussion to emphasize all tested regimens were well tolerated, although the more pronounced immunogenicity in individuals boosted with mRNA-1273 was associated with a higher percentage of individuals with local and systemic adverse events (**p. 16**).

Editorial Note: In the absence of reviewer 1 a mediating reviewer commented in their place:

Review of revised manuscript

Klemis et. al.

Head-to-Head-to-head analysis of immunogenicity and reactogenicity of heterologous ChAdOx1 nCoV-19-priming and BNT162b2 or mRNA-1273-boosting with homologous COVID-19 vaccine regimens

Reviewer #1 had 2 major concerns:

1. Discussion of prior randomized controlled trials and clearly noting the differences in prime-boost timing between those trials and this study.

A number of relevant publications are now referenced. The authors have clarified the nature of their cohort (convenience sample) and pointed out the differences in prime-boost interval.

2.Refrain from using “superior/inferior” descriptors and ranking vaccines due to effect this may have on vaccine hesitancy, especially in areas of the world without access to the ‘superior’ vaccines.

The authors revised the manuscript to include objective descriptions of differences between vaccines in terms of immunogenicity and/or reactogenicity. These edits are acceptable.

Minor concerns

The authors have responded to each of these appropriately.

Please check with authors about line 222. They wrote heterologous ‘BNT/ChAdOx’. The rest of the manuscript is careful to label groups in order of vaccine received. Did they mean ‘ChAdOx/BNT’?

Reviewer #2 concerns

The authors have responded to each of these appropriately.

REVIEWERS' COMMENTS

Reviewer #2 (Remarks to the Author):

The authors have appropriately addressed the reviewers concerns. The additional information and analyses strengthen the manuscript and increase its value to readers.